# *TSEN54* missense variant in Standard Schnauzers with leukodystrophy

**Theresa Störk**[1‡], **Jasmin Nessler**[2‡], **Linda Anderegg**[3‡], **Enrice Hünerfauth**[2], **Isabelle Schmutz**[3], **Vidhya Jagannathan**[3], **Kaisa Kyöstilä**[4,5,6], **Hannes Lohi**[4,5,6], **Wolfgang Baumgärtner**[1‡], **Andrea Tipold**[2‡], **Tosso Leeb**[3‡*]

**1** Department of Pathology, University of Veterinary Medicine Hannover, Foundation, Hannover, Germany, **2** Department for Small Animal Medicine and Surgery, University of Veterinary Medicine Hannover, Foundation, Hannover, Germany, **3** Institute of Genetics, Vetsuisse Faculty, University of Bern, Switzerland, **4** Department of Veterinary Biosciences, University of Helsinki, Helsinki, Finland, **5** Folkhälsan Research Center, Helsinki, Finland, **6** Department of Medical and Clinical Genetics, University of Helsinki, Helsinki, Finland

‡ TS, JN, and LA are joint first authors on this work. WB, AT, and TL are joint senior authors on this work.
* tosso.leeb@vetsuisse.unibe.ch

**Data Availability Statement:** SNV genotypes are given in the S1 File. Genome sequences accessions are given in the S3 Table.

## Abstract

We report a hereditary leukodystrophy in Standard Schnauzer puppies. Clinical signs occurred shortly after birth or started at an age of under 4 weeks and included apathy, dysphoric vocalization, hypermetric ataxia, intension tremor, head tilt, circling, proprioceptive deficits, seizures and ventral strabismus consistent with a diffuse intracranial lesion. Magnetic resonance imaging revealed a diffuse white matter disease without mass effect. Macroscopically, the cerebral white matter showed a gelatinous texture in the *centrum semiovale*. A mild hydrocephalus internus was noted. Histopathologically, a severe multifocal reduction of myelin formation and moderate diffuse edema without inflammation was detected leading to the diagnosis of leukodystrophy. Combined linkage analysis and homozygosity mapping in two related families delineated critical intervals of approximately 29 Mb. The comparison of whole genome sequence data of one affected Standard Schnauzer to 221 control genomes revealed a single private homozygous protein changing variant in the critical intervals, *TSEN54*:c.371G>A or p.(Gly124Asp). *TSEN54* encodes the tRNA splicing endonuclease subunit 54. In humans, several variants in *TSEN54* were reported to cause different types of pontocerebellar hypoplasia. The genotypes at the c.371G>A variant were perfectly associated with the leukodystrophy phenotype in 12 affected Standard Schnauzers and almost 1000 control dogs from different breeds. These results suggest that *TSEN54*:c.371G>A causes the leukodystrophy. The identification of a candidate causative variant enables genetic testing so that the unintentional breeding of affected Standard Schnauzers can be avoided in the future. Our findings extend the known genotype-phenotype correlation for *TSEN54* variants.

**Funding:** This study was supported by grants from the Albert-Heim Foundation (no. 105) to TL, Wisdom Health (HL), and the Jane and Aatos Erkko Foundation (HL). The funders had no role in study design, data collection and analysis, decision to publish, or preparation of the manuscript.

**Competing interests:** HL provides consultancy to Genoscoper Laboratories, which provides genetic testing for dogs.

## Author summary

Various hereditary diseases of the cerebral white matter occur in humans and dogs. We describe a new leukodystrophy in Standard Schnauzers. Genetic mapping and whole genome sequence analysis identified a likely candidate causative variant in the *TSEN54* gene encoding tRNA splicing endonuclease 54. These results provide new information about the role of *TSEN54* in cell metabolism and the development of the central nervous system in the late gestational and early post-natal period. The affected dogs potentially represent a translational large animal model for similar leukoencephalopathies in human medicine. The clinical phenotype in Schnauzers included multifocal central nervous system signs. A holistic pathogenically driven understanding of disease initiation and perpetuation requires a solid analysis of the underlying genetics and characterization of the disease phenotype at the clinical and cellular as well as sub-cellular level. In contrast to the canine phenotype with a predominant manifestation in the cerebrum white matter, other *TSEN54* variants in humans have been reported to result in a different pathological phenotype characterized by pontocerebellar hypoplasia. The differences between humans and dogs underscore the need for comparative analysis at the clinical, pathological and molecular level to understand species-specific protein mediated pathways, interactions and outcomes.

## Introduction

The term leukoencephalopathy refers to several disorders affecting the white matter of the central nervous system (CNS) [1,2]. In most cases oligodendrocytes are directly or indirectly affected by derangement of cellular and molecular pathways causing reduced myelin production consisting of diminished quantities, quality or both of this essential component [2]. Depending on the underlying pathology, leukoencephalopathies can be further divided into two major categories: leukodystrophy and hypomyelination.

In human medicine, hypomyelination, also known as hypomyelinogenesis or dysmyelinogenesis, is mostly associated with genetic, rarely viral or toxic disorders leading to insufficient or delayed formation of myelin [3,4]. On the other hand, the term leukodystrophy refers to progressive disorders of glial cells and myelin maintenance [1,5] resulting in bilateral symmetric lesions in selective areas of the CNS white matter [4]. The diagnosis is accomplished via a combination of the clinical course of the disease, magnetic resonance imaging (MRI), pathology and genetic testing [1,4]. Abnormal formation, turnover and destruction of the myelin are often caused by a lack of specific enzymes and inborn errors of metabolism [1,4,5].

In veterinary medicine, leukodystrophies were described in many different dog breeds, such as leukomyeloencephalopathy in Rottweiler [6–8] and Leonberger dogs [9], globoid cell leukodystrophy or Krabbe's disease in West Highland White Terriers [10,11] and Australian Kelpies [12], cavitating leukodystrophy in Dalmatians [13], fibrinoid leukodystrophy or Alexander's disease in a Labrador Retriever [14] and necrotizing myelopathy in Afghan hounds [15]. In some of these diseases an autosomal recessive mode of inheritance was described [11–13]. Causative genetic variants have been identified in *GFAP* (Labrador Retriever, [14]), *NAPELD* (Leonberger Dogs, [9]), and *VPS11* (Rottweiler dogs, [8]). The present study aimed to characterize clinical and pathological features of a new leukodystrophy in Standard Schnauzer puppies and to identify its underlying genetic cause.

## Results

### Clinical description

A dog breeder reported neurological deficits affecting multiple Standard Schnauzer puppies over the last ~10 years. Several puppies from different litters involving different dams were weak and showed progressive neurological signs such as dysphagia, non-ambulatory tetraparesis or sudden death.

For further evaluation of a potential genetic defect in Standard Schnauzers, six puppies (no. 1–4 and 13–14) of two different litters, 4 weeks of age, and one mother of the litters (no. 15) were presented to the Neurology Service of the Department for Small Animal Medicine and Surgery, University of Veterinary Medicine Hannover.

Two of the six presented puppies (no. 13 and 14) and the bitch (no. 15) were clinically unremarkable (S1 Table). The remaining four puppies (no. 1–4) were smaller than the unaffected siblings (1.0–1.4 kg versus 1.8–2.0 kg). Clinical signs included hypermetric ataxia, circling, dysphoria, head tilt (ipsi- or contralateral to direction of circling), bilateral ventro-lateral strabismus and generalized tonic-clonic seizures at the neurological examination. Neuroanatomical localization in affected puppies indicated diffuse intracranial lesions with a predominance of forebrain signs.

Basic clinical pathology (differential cell count, liver enzymes, bile acid, bilirubin, urea, creatinine, glucose, total protein, albumin, and electrolytes) of all puppies and cytological examination of the bitch´s milk were unremarkable.

Magnetic resonance imaging (MRI) of the brain of one affected puppy (no. 1) revealed mildly enlarged lateral ventricles and a well demarcated bilateral almost symmetrical lesion affecting the deep and subcortical white matter of the cerebrum diffusely without any mass effect (Fig 1A–1D). The lesion was markedly hyperintense in T2-weighted (T2w) images and hypointense in T1-weighted (T1w) and fluid accentuated inversion recovery sequence (FLAIR). Partial loss of cerebral parenchyma led to a higher content of cerebrospinal fluid (CSF) within the tissue. No pathological contrast enhancement was visible. MRI of one of the unaffected Schnauzer puppies (no. 14) was performed to exclude subclinical lesions and revealed a decreased distinction between white and grey matter (Fig 1E–1H) indicating a premature myelination which is considered normal at the age of 4 weeks [16]. CSF could not be obtained. Due to the severe and progressive clinical signs, the four affected puppies were humanely euthanized.

### Necropsy and light microscopy

A total of 12 Standard Schnauzer puppies with neurological disease were examined (no. 1-12). These included the four clinically investigated puppies (no. 1–4). Relevant pathomorphological findings during necropsy were restricted to the CNS. The cerebrum of all 12 puppies showed macroscopically a mild bilateral internal hydrocephalus and the borders between gray and white matter were indistinct. Additionally, the white matter of the cerebrum had a gelatinous consistence and showed edema in the 4 week old puppies no. 1–4.

Histologically, a mild (no. 5–12) to severe (no. 1–4), bilateral symmetric multifocal lack of myelin, mainly in the *centrum semiovale* was observed using hematoxylin and eosin (HE) and luxol fast blue and cresyl echt violet (LFB) staining. Additionally, a moderate, diffuse edema was found in the white matter of the cerebrum of affected animals no. 1–4 (Figs 2–4).

Using β-amyloid precursor protein (APP) specific immunohistochemistry, mild, multifocal axonal damage was noticed in areas of severe lack of myelin (Fig 4C and 4D). In the combined

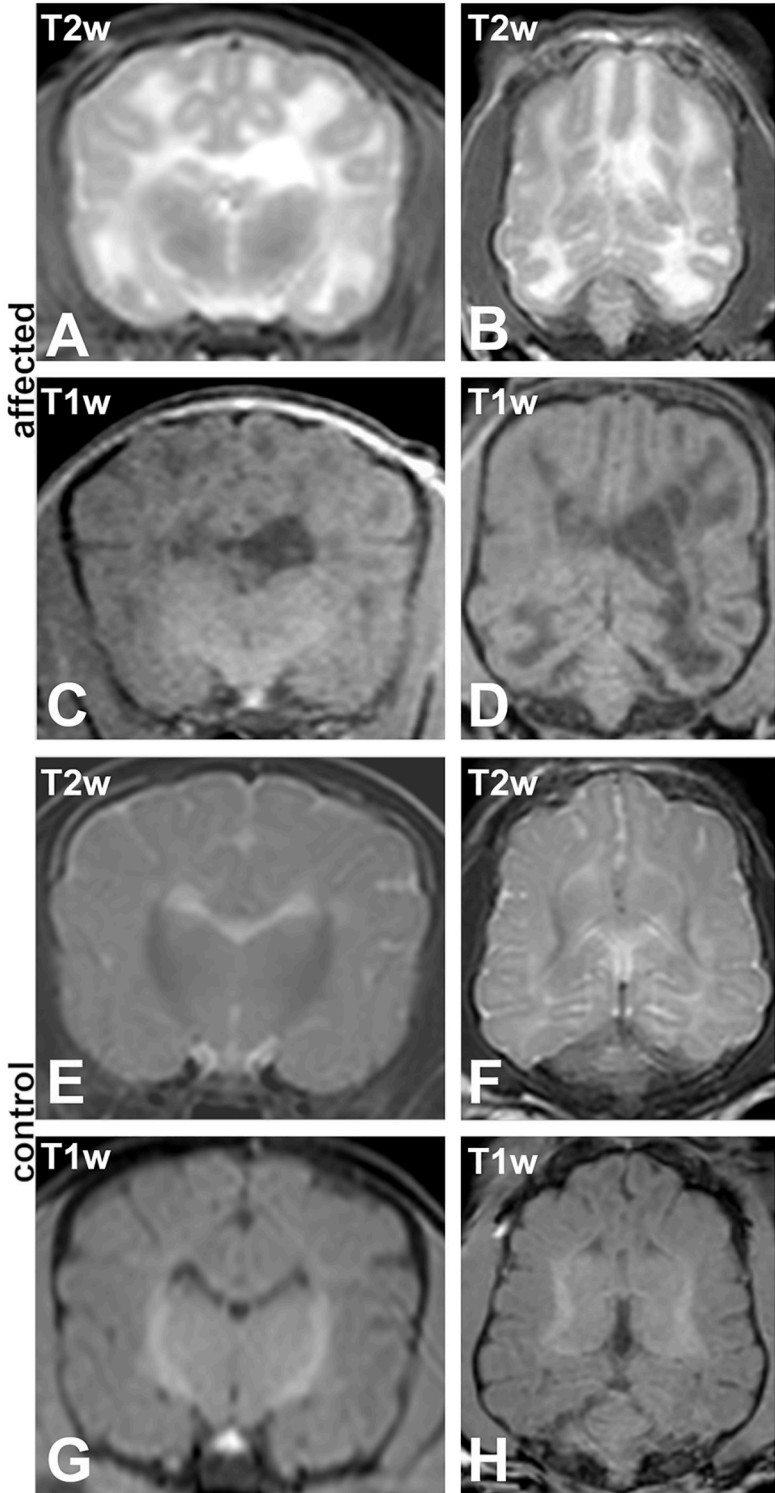

**Fig 1. Magnetic resonance imaging (MRI).** (A-D) Affected 4 weeks old Schnauzer puppy (no. 1) and (E-H) a clinically normal sibling at the same age (no. 14). (A, B, E, F) T2 weighted (T2w) sequences; (A, E) transversal and (B, F) dorsal. (C, D, G, H) T1 weighted (T1w) sequences; (C, G) transversal and (D, H) dorsal. MRI reveals mildly enlarged and asymmetric lateral ventricles and a T2w-hyperintense/T1w-hypointense lesion diffusely involving the whole cerebral white matter in affected animals. The age matched unaffected sibling shows beginning myelination in the central deep white matter, typical for the age.

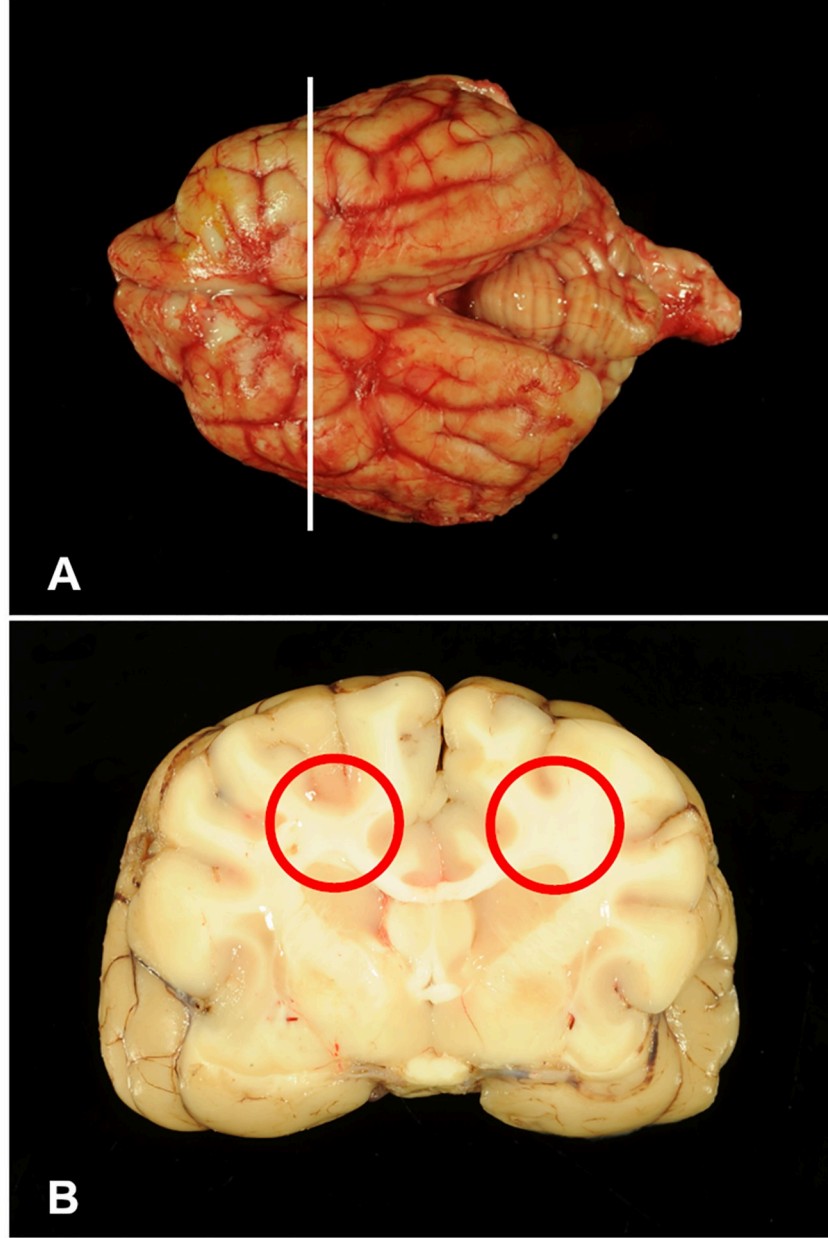

**Fig 2. Macroscopic overview.** (A, B) Age matched healthy control no. 1009. (A) Dorsal view on cerebrum, cerebellum and brainstem with section plane (white line). (B) Cross section of cerebrum showing the *centrum semiovale* (red circle).

APP immunohistochemistry and LFB staining the colocalization of axonal damage and myelin loss was visualized (Fig 4E and 4F).

In addition to the more general histochemical stain for myelin, LFB, two other different myelin stains were applied, using 2′,3′-cyclic nucleotide-3′-phosphodiesterase (CNPase) as a marker for early myelin formation and myelin basic protein (MBP) as the prototype of mature myelin formation.

CNPase immunohistochemistry revealed almost no detectable myelin within the *centrum semiovale* and a slightly reduced number of oligodendrocytes in puppies no. 1–4. However, the

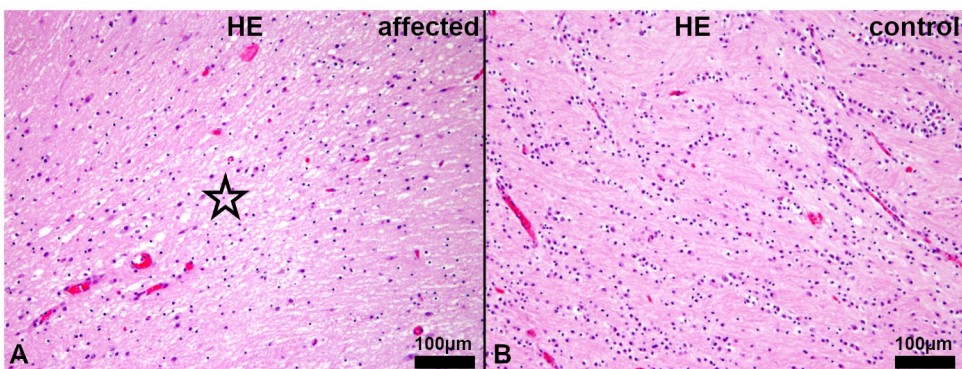

**Fig 3. Histopathology of the *centrum semiovale* of the cerebrum.** (A) Affected puppy no. 1 and (B) age matched healthy control no. 1009. (A) Reduced eosinophilia and moderate, diffuse edema of the white matter (asterisk) of affected puppy. (HE). (B) Regular diffuse eosinophilia, indicating normal myelination, in the white matter of an age matched control puppy. (HE).

cytoplasm of the oligodendrocytes and their fragmented processes were strongly positive (Fig 5A and 5B). Reduced myelination within *centrum semiovale* was confirmed using an anti-MBP antibody. Only single, fine strands of myelin were detected in the affected CNS regions in contrast to the dense meshwork of myelin in unaffected areas of the same animal and in age matched controls no. 1009–1016 (Fig 5C and 5D).

We quantified mature myelinating oligodendrocytes using an antibody against neurite outgrowth inhibitor A (NogoA). Within the *centrum semiovale*, affected Schnauzer puppies show only single NogoA$^+$ oligodendrocytes, whereas age matched healthy control animals show up to eight times as many myelinating NogoA$^+$ oligodendrocytes as the affected animals in this area (Fig 6).

Glial fibrillary acidic protein (GFAP)-immunohistochemistry detected a mildly reduced density of astrocytes within the white matter. Most of these astrocytes had a plump morphology and developed shorter and reduced numbers of processes in comparison to the age matched controls (Fig 7A and 7B). Additionally, using the antibody against ionized calcium binding adaptor molecule 1 (Iba-1) a higher number of macrophages/microglia within the affected white matter was detected, whereas an equal distribution of microglia within white and grey matter was observed in in the age matched controls. The majority of the macrophages/microglia within the white matter had an amoeboid morphology (Fig 7C and 7D). Cerebellum, brain stem, spinal cord and peripheral nerves were macroscopically and histologically unremarkable in the affected dogs.

## Genetic analysis

The occurrence of leukodystrophy in multiple consecutive litters from related parents suggested an inherited disease. Pedigree analysis of the four affected puppies no. 1–4 revealed several inbreeding loops and was suggestive for a monogenic autosomal recessive inheritance (Fig 8). We obtained genomic DNA samples from the four cases (no. 1–4), five unaffected littermates (no. 13, 14, 19–21) and the four unaffected parents (no. 15–18) of these two litters. Linkage and autozygosity analysis were performed in the 13 available samples. Eight genomic segments on six different chromosomes with a total of 29,237,328 bp or roughly 1.2% of the 2.4 Gb dog genome showed linkage in both litters and shared homozygous genotypes in the four available cases (S2 Table).

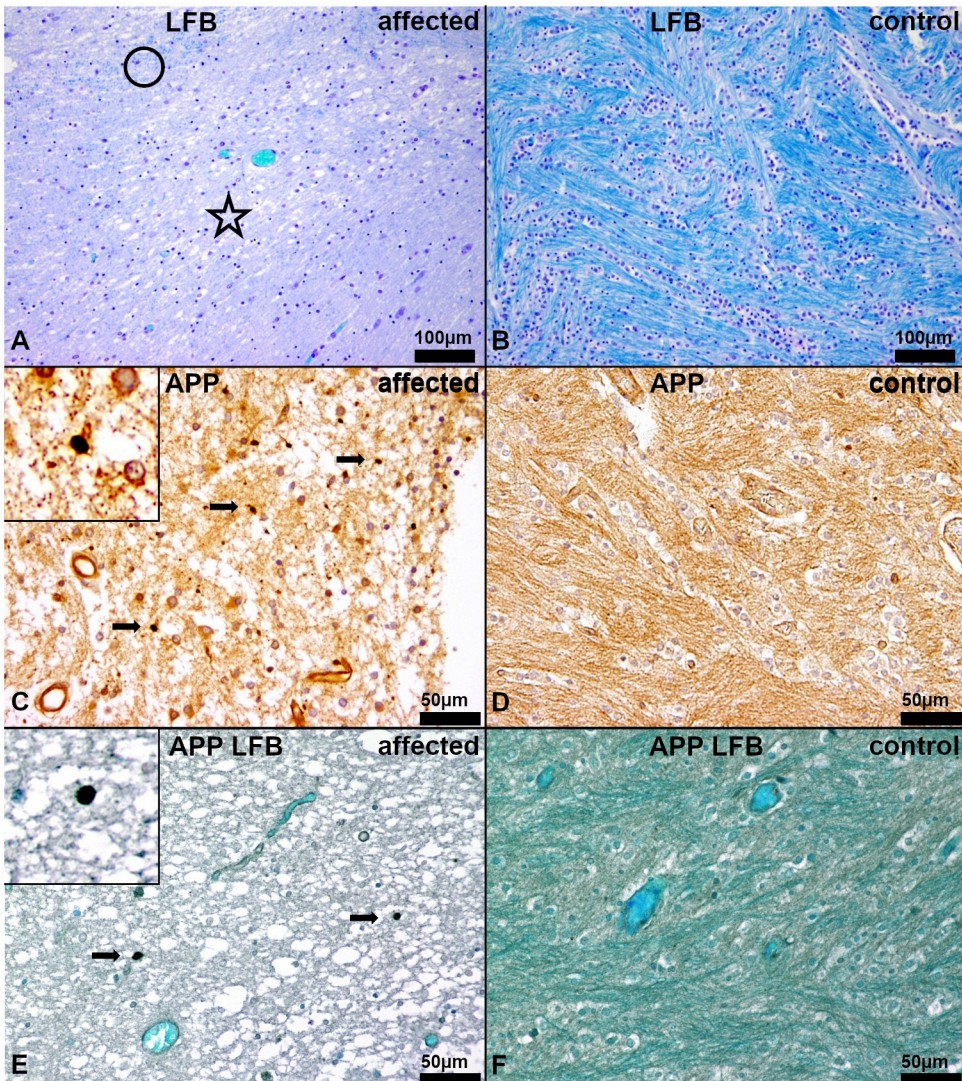

**Fig 4. Histochemistry and immunohistochemistry of the white matter in the *centrum semiovale* of the cerebrum.**
(A, C, E) Affected puppy no. 1 and (B, D, F) age matched healthy control no. 1009. (A) Severe, diffuse lacking bluish staining, indicating myelin loss and edema in the *centrum semiovale* (asterisk). Only single fine strands of myelin can be detected (circle). (LFB). (B) Regularly developed myelin characterized by prominent bluish staining in the white matter (dark blue). (LFB). (C) Areas of severe myelin loss in the diseased puppies reveal low numbers of damaged axons (arrows). Vacuolization and loosening of the parenchyma indicate moderate edema. Inset shows a damaged axon at a higher magnification. (β-amyloid precursor protein, APP). (D) No axonal damage is detectable in healthy control animals. (APP). (E) Severe, diffuse lacking blue-greenish staining, indicating myelin loss and edema, in the *centrum semiovale*, containing low numbers of damaged axons (arrows). Vacuolization and loosening of the parenchyma represent the moderate edema. Inset shows a damaged axon at a higher magnification. (APP, LFB). (F) Regularly developed myelin characterized by prominent blue-greenish staining in the white matter (dark teal) without any damaged axons. (APP, LFB).

Whole genome sequencing of one of the affected dogs was performed (no. 2). Private homozygous protein-changing variants were identified by comparing the variants in the case with the genomes of 8 wolves and 213 dogs from various breeds (Table 1, S3 Table). The potential leukodystrophy status of control animals was unknown. However, as this is a rare and severe condition, we assumed the control dogs and wolves to be homozygous for the genome reference allele at the causative variant.

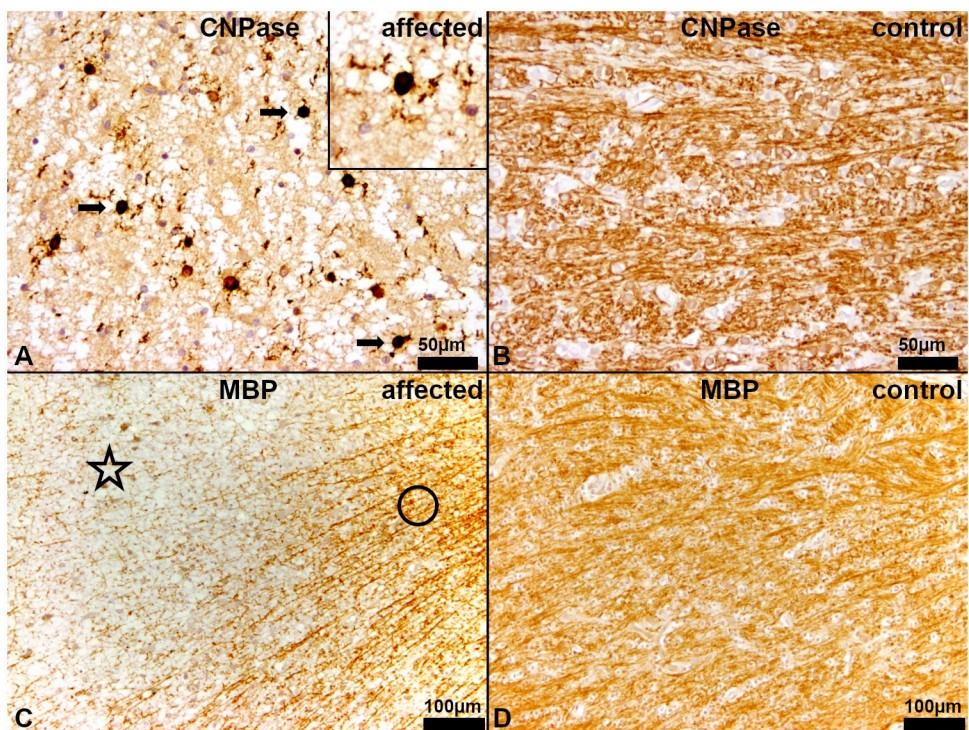

**Fig 5. Immunohistochemistry of the white matter in the *centrum semiovale* of the cerebrum.** (A, C) Affected puppy no. 1 and (B, D) age matched healthy control no. 1009. (A) In affected puppies almost no myelin is detectable whereas oligodendrocytes show fragmented processes and a strongly perinuclear cytoplasmic positive signal (arrows). Inset shows an oligodendrocyte with strong perinuclear cytoplasmic signal in a higher magnification. (CNPase). (B) In age matched controls the white matter shows a marked, diffuse brownish staining of the myelin. (CNPase). (C) Only single, fine strands of myelin (asterisk) were detected in contrast to the dense meshwork of myelin in adjacent non-affected areas (circle) of diseased puppies. (MBP). (D) Age matched control puppies show a diffuse staining of the myelin of the white matter (MBP).

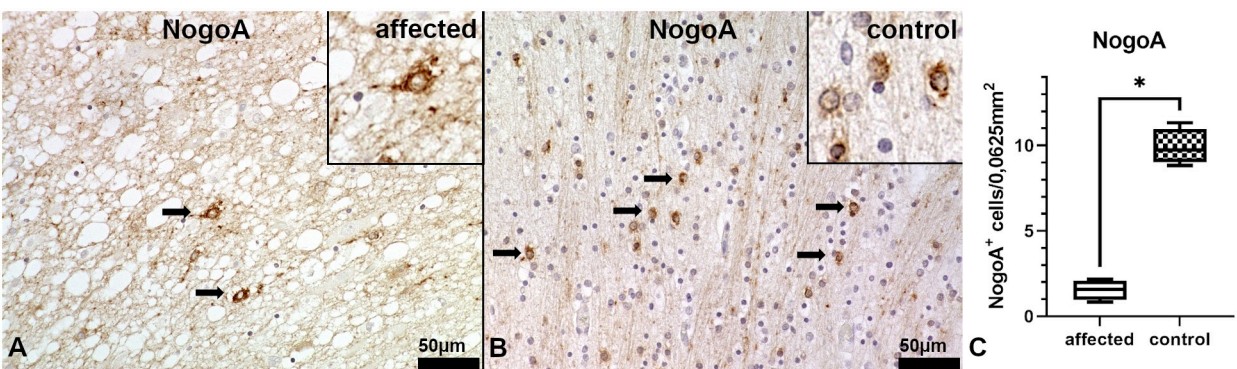

**Fig 6. Immunohistochemistry of the white matter in the *centrum semiovale* of the cerebrum.** (A) Affected puppy no. 1 and (B) age matched healthy control no. 1009. (A) In affected puppies, only single NogoA+ oligodendrocytes are detectable (arrows). Vacuolization and loosening of the parenchyma indicate the moderate edema. Inset shows a NogoA+ oligodendrocyte at a higher magnification. (NogoA). (B) Age matched control animals show numerous NogoA+ myelinating oligodendrocytes. Inset shows NogoA+ oligodendrocytes at a higher magnification. (NogoA). (C) Statistical analysis of NogoA-immunohistochemstry shows significantly decreased numbers of myelinating oligodendrocytes in the *centrum semiovale* of the affected Schnauzer puppies (n = 4, no.1, 2, 3 and 4) in comparison to age matched control animals (n = 4, no. 1009, 1010, 1011 and 1012). Box and whisker plots display median and quartiles with maximum and minimum values. Significant difference ($p \leq 0.05$, Mann-Whitney U-test) is labeled by asterisk.

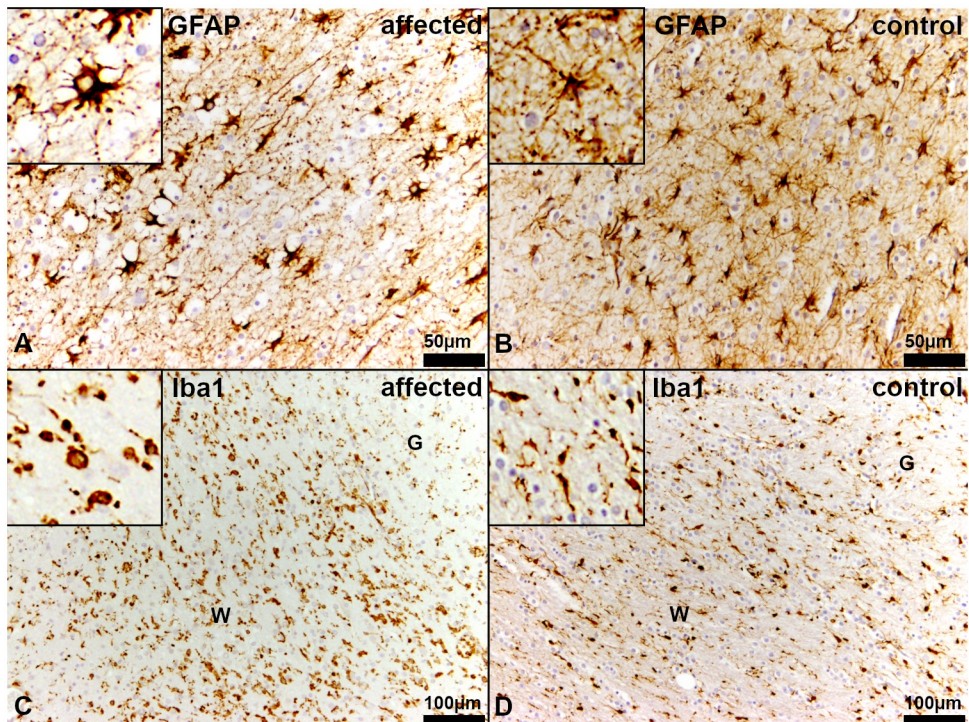

**Fig 7. Immunohistochemistry of the white matter in the *centrum semiovale* of the cerebrum.** (A, C) Affected puppy no. 1 and (B, D) age matched healthy control no. 1009. (A) There is a reduced density of astrocytes in the white matter of affected puppies. Most of the astrocytes have a plump morphology developing few short and bulky processes. Inset shows an astrocyte at a higher magnification. (GFAP). (B) In the white matter of age matched control puppies, astrocytes have long, slender branching processes. Inset shows an astrocyte at a higher magnification. (GFAP). (C) Diseased Schnauzer puppies reveal an elevated number of macrophages/microglia in the white matter. Many of these macrophages/microglia have an amoeboid or reactive morphology (arrows). Inset shows amoeboid macrophages/microglia at higher magnification. (Iba1). (D) In healthy control dogs, there is an equal distribution of microglia in the white (W) and the grey (G) matter. Most of the macrophages/microglia represent the ramified, non-reactive type. Inset shows macrophages/microglia at a higher magnification. (Iba1).

The variant calling pipeline detected 2.9 million homozygous variants in the genome of the sequenced case. Of these, 32 were absent from the control genomes and predicted to be protein-changing. Only one private homozygous protein-changing variant was located in the critical intervals of combined linkage and autozygosity (Table 1, S4 Table). The variant can be designated as Chr9:5,015,506C>T (CanFam 3.1 assembly). It was a missense variant affecting exon 5 of the *TSEN54* gene, XM_540434.6:c.371G>A, and predicted to result in a non-conservative exchange of a glycine into an aspartic acid, XP_540434.3:p.(Gly124Asp). The wildtype glycine is conserved in TSEN54 orthologs across phylogenetically diverse vertebrates (Fig 9).

We genotyped the variant in a larger cohort of 381 Standard Schnauzers (no. 1–381). This cohort included the four cases from the linkage and autozygosity mapping experiment (no. 1-4) and an additional eight histopathologically confirmed cases from which only formalin-fixed material was available (no. 5–12). The genotypes at the *TSEN54*:c.371G>A variant were perfectly associated with the phenotype ($P_{Fisher} = 6.1 \times 10^{-22}$; Table 2). All 12 affected dogs carried the variant in homozygous state. Among the other 369 Standard Schnauzers, we observed 344 dogs that were homozygous wildtype (93.2%) and 25 dogs that were heterozygous (6.8%) and presumably carriers for the disease. The sires and dams (no. 15–18) of the four affected dogs from these two litters were heterozygous (obligate carriers). Most carriers of this variant were close relatives of the affected Standard Schnauzer puppies. Our cohort included 237 Standard

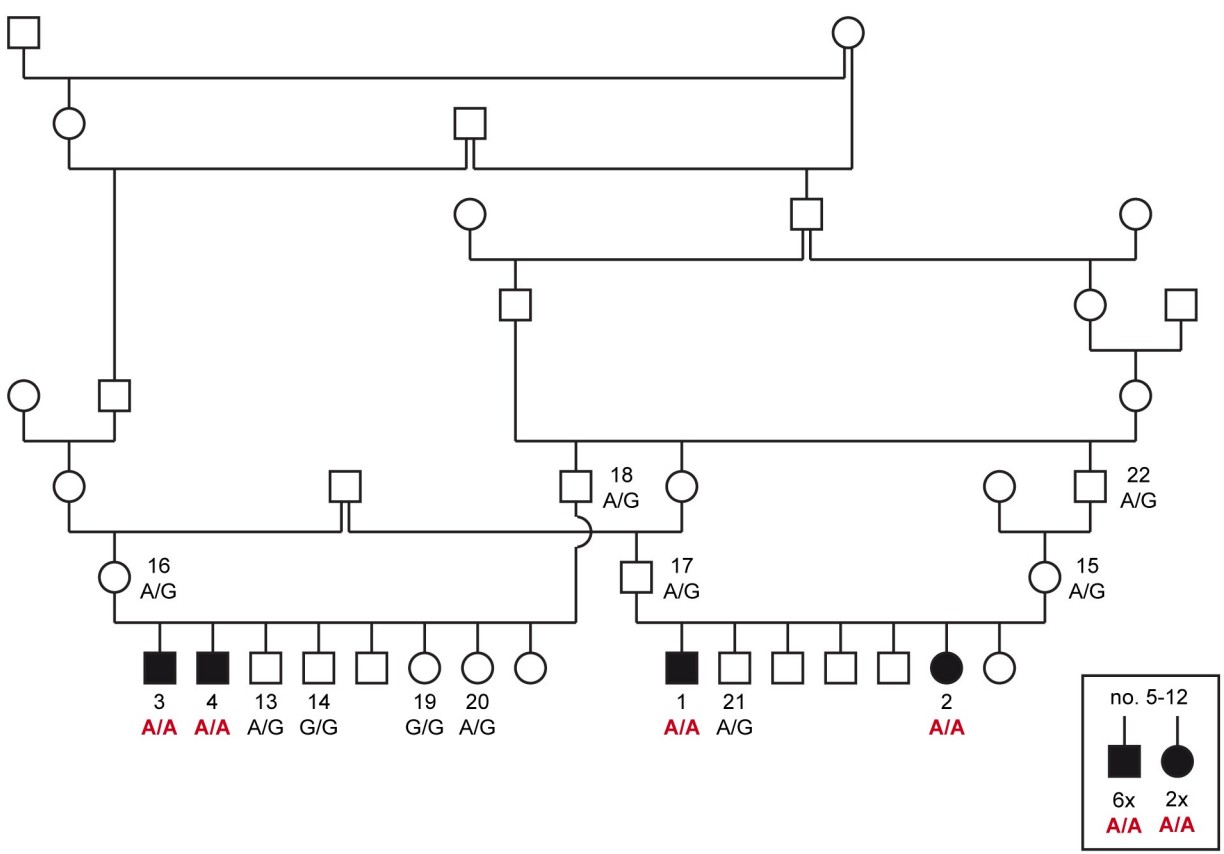

**Fig 8. Pedigree of Standard Schnauzers used for linkage analysis and autozygosity mapping.** Filled symbols represent dogs with leukodystrophy. For a better overview, the pedigree was drawn only with a subset of the closely related dogs. Two litters with a total of four affected dogs and their parents were used for linkage analysis. The four cases no. 1–4 were used for autozygosity mapping. Numbers indicate dogs, from which DNA samples were available. The solitary symbols in the square represent 8 additional affected Standard Schnauzers, for which no pedigree data was available (no. 5–12). For these dogs, only FFPE material was available for DNA isolation. The resulting DNA was of insufficient quality for genotyping on microarrays, but could be used to determine the genotypes at the *TSEN54*:c.371G>A variant. The *TSEN54*:c.371G>A genotypes for all dogs with DNA samples are indicated in the figure.

Schnauzers from Finland, which were not specifically collected for this study. In this subset of dogs, there was only one carrier suggesting that the disease allele may not yet be widespread globally. We also genotyped 627 dogs from 72 genetically diverse breeds. None of these dogs carried the mutant allele at the *TSEN54*:c.371G>A variant (S5 Table).

**Table 1. Variants detected by whole genome resequencing of an affected Standard Schnauzer.**

| Filtering step | Variants[a] |
|---|---:|
| Homozygous variants in whole genome | 2,891,805 |
| Private homozygous variants in the whole genome[b] | 5,258 |
| Private homozygous protein changing variants in the whole genome | 32 |
| Private homozygous protein changing variants in critical intervals[c] | 1 |

[a] All called variants were counted (no quality filter was used).

[b] Private variants were exclusively called in the affected dog and had homozygous reference or missing genotype calls in 221 control genomes.

[c] Intervals of combined linkage and homozygosity (S2 Table)

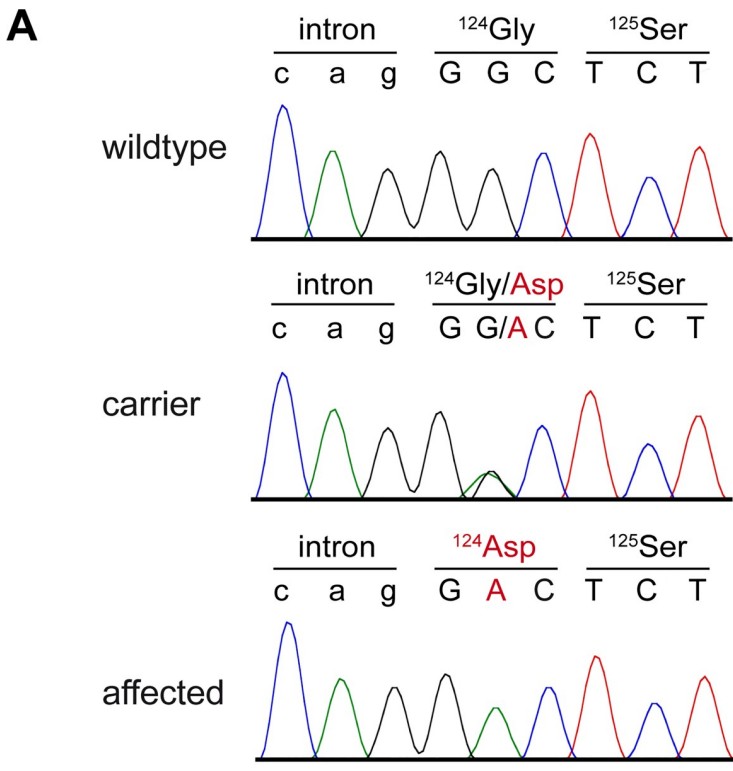

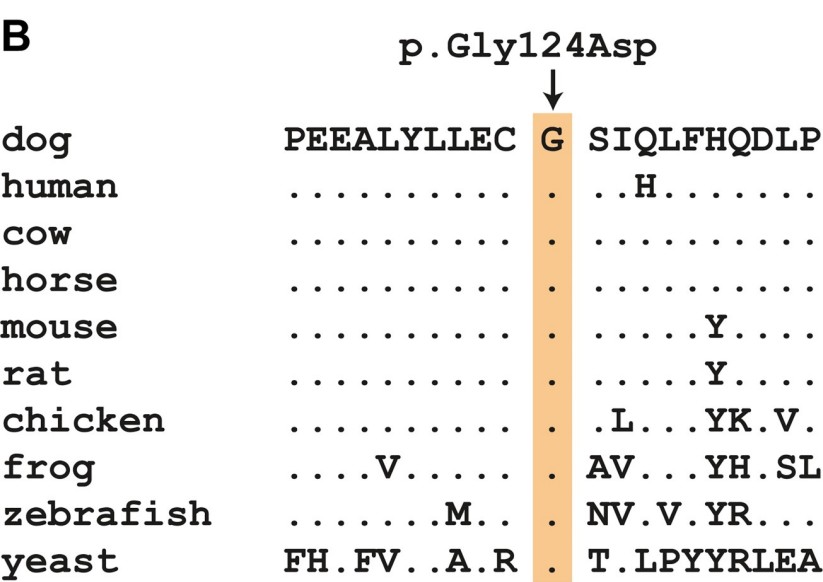

**Fig 9. Details of the *TSEN54*:c.371G>A variant.** (A) Sanger electropherograms from dogs with the three different genotypes confirm the presence of the variant. (B) A multiple species alignment of the TSEN54 amino acid sequence illustrates that the variant affects a highly conserved region of the protein. The wildtype glycine is perfectly conserved across diverse vertebrates. The yeast (*S. cerevisiae*) SEN54 protein shows overall low amino acid sequence homology with the vertebrate TSEN54 ortholog. However, the glycine residue is conserved and sits within a structurally conserved region consisting of an alpha-helix followed by a beta-sheet.

**Table 2. Association of the *TSEN54*:c.371G>A genotypes with leukodystrophy.**

| Genotype | G/G | A/G | A/A |
|---|---|---|---|
| Affected Standard Schnauzers (n = 12) | - | - | 12 |
| Standard Schnauzers, population controls (n = 369) | 344 | 25 | - |
| Dogs from other breeds (n = 627)[a] | 627 | - | - |

[a]These dogs do not include the 221 control genomes, which were used for the initial analysis. Thus, in total, the mutant allele was absent from >800 dogs of other breeds.

Immunofluorescence of the cerebrum with an anti-TSEN54 antibody revealed a similar intra-nucleolar signal in both affected and control animals (no. 1–4 vs. 1009–1016). The distribution and intensity of the signals were equal between cases and controls (Fig 10).

## Discussion

In this study, we describe a new inherited leukodystrophy in dogs and identify the TSEN54:c. Gly124Asp missense variant as a candidate cause of the disease.

The progressive clinical course, the lesions visible in MRI and loss of myelin in the cerebrum seen light microscopically point towards a demyelinating leukodystrophy rather than to a hypomyelinating disease. Although there are many different types of leukodystrophy described in dogs [6–15], the distribution and also the histopathological appearance of the lesions described in the *TSEN54* mutant dogs were different compared to previously described cases of canine leukodystrophy.

Leukoencephalopathies are a heterogeneous group of diseases affecting the white matter of the brain. The etiology as well as clinical signs, findings in diagnostic imaging and histopathology are highly variable. In most leukoencephalopathies oligodendrocytes and myelin are predominantly affected, whereas axonal damage represents a rather infrequent finding. However, axonal damage in areas of myelin lack as shown in the present cases, has been reported as an epiphenomenon in leukodystrophies [17]. In several leukodystrophic diseases axonal damage is not directly caused by the lack of myelin or oligodendrocytes and rather represents a secondary event [18]. Several possible mechanisms of axonal damage in white matter diseases, such as sulfatide storage in metachromatic leukodystrophy in humans or aberrant glutamate homeostasis and nitric oxide production in multiple sclerosis have been described [19]. In addition, an impairment of axons as a consequence of demyelination and the associated loss of myelin-

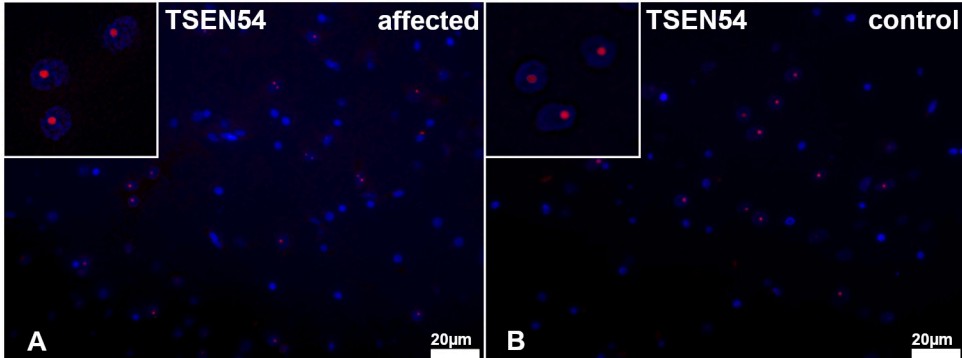

**Fig 10. TSEN54 immunofluorescence of the white matter in the *centrum semiovale* of the cerebrum.** (A) Affected puppy no. 1 and (B) age matched healthy control no. 1009. There is an almost equal number and distribution of cells showing an intranuclear positive signal for TSEN54 (red). Insets show TSEN54[+] cells in a higher magnification.

derived trophic support and ion imbalance according to the outside-in model cannot be excluded [20]. The pathomechanism of axonal damage in *TSEN54* associated leukodystrophy is still unclear and warrants additional investigations.

An inborn lack of central myelination leads to generalized tremor as observed in Springer Spaniels, Weimaraner dogs or Chow Chows [21–23]. Clinical signs, such as whole body tremor are present shortly after birth, might remain unchanged or decrease in severity with time [21–23]. MRI findings in hypomyelination syndromes are rather subtle and include mildly altered signal intensity of the white matter in comparison to age matched controls and resembles the myelin pattern of newborn individuals [4].

Destruction of initially normal myelin in the course of the disease is a characteristic feature of leukodystrophies [2]. The results of the different myelin stains, underline the leukodystrophic character of the disease of the present cases [24,25]. CNPase immunohistochemistry detecting both immature oligodendrocytes and early myelin formation showed the presence of oligodendrocytes and the predominant absence of myelin in the *centrum semiovale* of the affected animals. In addition, the MBP immunohistochemistry revealed single fine strands of mature myelin in affected areas. This could be interpreted as an insufficient and inadequate myelin formation mediated by the detected genetic defect. Affected puppies are initially normal and develop progressive neurological signs mostly at 1–6 months of age dependent on the localization of the lesions. In humans, a systematic analysis of MRI patterns is used to differentiate leukodystrophies from other neurological diseases in children [4]. In veterinary medicine, MRI findings in leukodystrophies were only described in very few cases so far. Usually marked T2w hyperintense lesions are visible bilateral symmetrically in the white matter mostly of the forebrain without any mass effect. Depending on the disease, different distribution patterns of the lesions were described, affecting mostly the subcortical or the deep cerebral white matter with or without formation of cavitation [10,26,27]. It is very important to include all available information concerning macroscopy, histopathologic lesions, anamnesis and also genetics to come to an appropriate diagnosis.

Several genetic defects have previously been associated with leukodystrophies. In this study, we identified the *TSEN54*:c.371G>A missense variant in affected dogs. The variant was identified during the analysis of two closely related litters with four affected puppies. A combination of linkage analysis, autozygosity mapping, and whole genome sequencing of one of the cases identified this variant as the only protein-changing sequence variant that was located in the critical intervals and private to the sequenced case. The mapping of the disease locus delineated several very large genomic intervals comprising more than 29 Mb. Therefore, the results from the two litters suggested *TSEN54*:c.371G>A as candidate causative variant with only relatively weak statistical support. During the further course of the research project, DNA from archived FFPE material of eight additional cases became available for genetic analyses. Considering that the mutant A-allele is overall relatively rare in Standard Schnauzers, the homozygous mutant A/A genotype in these eight additional cases provides very strong additional support for a presumed causative role of the *TSEN54*:c.371G>A missense variant in the phenotype. Taken together, all 12 Standard Schnauzers with leukodystrophy were homozygous A/A, while this genotype was not observed in ~1000 other dogs.

On the other hand, it must be cautioned that the eight additional cases came from the same breeding lines as the initial four cases and may have been closely related. Thus, while the statistical support for the causality of the *TSEN54*:c.371G>A missense variant is very strong, it does not provide definitive proof for the causality. We cannot formally exclude the possibility that the *TSEN54*:c.371G>A variant is functionally neutral and in strong linkage disequilibrium to the true causative variant, which in this case might be e.g. a large structural variant and/or a non-coding regulatory variant. Large structural variants would have most likely been missed

by our variant calling pipeline and non-coding regulatory variants would have been filtered during our prioritization of protein-changing variants.

TSEN54 is a subunit of the tRNA splicing endonuclease complex [28]. Studies in yeast suggest that this complex consists of four subunits (SEN2, SEN34, SEN15 and SEN54) which play a role in multiple RNA processing events [29]. The exact three-dimensional structure of the eukaryotic tRNA splicing endonuclease complex has not yet been solved. The large SEN54 subunit is responsible for recognizing the precursor tRNA substrate and positioning the two catalytical subunits SEN2 and SEN34 to their respective cleavage sites [28]. In line with this role, yeast SEN54 is a highly basic protein, which facilitates its interaction with the negatively charged phosphates of the precursor-tRNA. The vertebrate ortholog of yeast SEN54, called TSEN54, shows relatively little sequence homology, but several structurally conserved motifs including a predicted alpha-helix followed by a beta-sheet at amino acids 117–131 [29]. The p.Gly124Asp variant in dogs with leukodystrophy replaces a glycine residue that is conserved across diverse eukaryotes from yeast to humans with a negatively charged aspartic acid. It is conceivable that either the spatial requirement of the side chain of the mutant aspartic acid interferes with the secondary structure of TSEN54 and/or that the additional negative charge hinders the binding of TSEN54 to the negatively charged precursor tRNA.

TSEN54 has not been directly associated with leukodystrophy before. In humans, high expression of TSEN54 has been detected in neurons of the pons, cerebellar dentate and olivary nuclei during the second trimester of pregnancy. A functional endonuclease complex is essential for the development of these regions [30]. Genetic variants in *TSEN54* cause pontocerebellar hypoplasia (PCH) type 2, 4 or 5 in humans [30–32]. PCH is characterized by hypoplasia of the cerebellum and the ventral pons, early postnatal death and progressive microcephaly and ventriculomegaly [30–33]. It is histopathologically associated with diffuse gliosis of the white matter in the brain [34,35]. In one human patient with PCH2, TSEN54:p.Tyr119Glu was identified as pathogenic variant. This variant is in close proximity to the canine p.Gly124Asp variant and also introduces an additional negative charge [36].

In zebrafish, knockdown of *TSEN54* results in brain hypoplasia and loss of structural definition in the brain due to increased cell death via loss of functional mechanism [37]. This could explain the phenotype found in the Standard Schnauzers with mutant TSEN54 described in our study. Dysfunction and early cell death of oligodendrocytes may lead to dysmyelination and impaired myelin maintenance which manifests as leukodystrophy. Immunohistochemical detection of TSEN54 in brains of affected Standard Schnauzers similarly to controls indicated a functional rather than a quantitative deficiency of TSEN54.

The fact that *TSEN54* variants mainly cause pontocerebellar hypoplasia in humans but leukodystrophy in the cerebrum of affected dogs might be explained by a possible distinct quality or degree of protein malfunction due to the different genetic variants within the gene. Alternatively, *TSEN54* might be of minor importance in the neuronal development of dogs compared to humans and instead be mainly involved in oligodendrocyte function in the canine species compared to others.

We describe here the first potentially pathogenic *TSEN54* variant in dogs. Unlike in humans, where variants in *TSEN54* cause pontocerebellar hypoplasia, the main finding in affected Standard Schnauzers is leukodystrophy. These findings expand the known genotype-phenotype correlation for *TSEN54* variants.

## Conclusions

This study indicates that *TSEN54* should be considered as a new candidate gene for leukodystrophy. Our findings enable genetic testing for Standard Schnauzers, which can be used to

avoid the unintentional breeding of affected puppies. This canine leukodystrophy might serve as a translational large animal model to better understand the function of the TSEN54 protein and its role in pathophysiology in health and disease.

## Materials and methods

### Ethics statement

All examinations were performed with written informed owner´s consent according to ethical guidelines of the University of Veterinary Medicine Hannover, Foundation. Blood samples were collected with the approval of the *Cantonal Committee for Animal Experiments* (Canton of Bern; permit BE75/16). Sample collection in Finland was ethically approved by the Animal Ethics Committee of State Provincial Office of Southern Finland (ESAVI/343/04.10.07/2016). All animal experiments were done in accordance with local laws and regulations.

### Clinical examinations

A breeder of Standard Schnauzers reported neurological deficits affecting multiple puppies from approximately ~2014 to ~2017. Several puppies from different litters and bitches were weak and showed progressive neurological signs such as dysphagia, non-ambulatory tetrapar-esis or sudden death. For further evaluation of a potential genetic defect in Standard Schnau-zers, six puppies of two different litters (no. 1–4, 13, 14) were presented to the Neurology Service of the Department for Small Animal Medicine and Surgery, University of Veterinary Medicine Hannover. One of the two dams (no. 15) of the affected puppies was also presented for clinical examination. Clinical examination of the puppies no. 1–4, 13–14 and one mother (no. 15) was performed by a resident and a Diplomate of the European College of Veterinary Neurology (ECVN). Details on each individual dog are summarized in the S1 Table.

### Magnetic resonance imaging

MRI of puppies no. 1 and 14 were performed in general anesthesia. Puppies were sedated with diazepam (0.5 mg/kg intravenously (i.v.), Diazepam lipuro, B. Braun Melsungen AG, Melsun-gen, Germany) and levomethadon (0.2 mg/kg i.v., L-Polamivet, MSD Animal Health GmbH, Unterschleißheim, Germany). General anesthesia was induced with propofol dosed to effect (1–3 mg/kg i.v., Narcofol, CP-Pharma, Burgdorf, Germany). Puppies were orotracheally intu-bated and connected to a semiclosed circle absorber system (Anaesthesia ventilator, Cato, Dräger, Langenhagen, Germany). Maintenance of anesthesia was performed with isoflurane (CP-Pharma) in an oxygen/air mixture (1:1, flow 50 ml/kg/min).

The following MRI sequences were obtained with a 3.0-T scanner (Achieva, Philips Medical Systems, Best, The Netherlands): T2w and T1w pre- and post-contrast administration (0.5 ml/ kg i.v., Dotarem, medithek GmbH, Oststeinbek, Germany) each in sagittal, transversal and dorsal section as well as transversal section of FLAIR.

### Necropsy and histological examination

Four euthanized Standard Schnauzer puppies (no. 1–4) were submitted for complete necropsy. Additionally, between 2014 and 2017, eight younger Standard Schnauzer puppies from the same breeder (no. 5–12), 3 to 21 days of age, had been sent to the Department of Pathology, University of Veterinary Medicine, Hannover, Germany, for post mortem examination. Path-ological examination was performed according to a routine protocol at the Department of Pathology, University of Veterinary Medicine, Hannover. During necropsy representative samples of all organs and tissues were taken and subsequently fixed in 10% neutral buffered

formalin before being embedded in paraffin wax. For histological examination, 2–3 μm thick sections were cut and stained with hematoxylin and eosin (HE). Additionally, luxol fast blue- and cresyl echt violet staining (LFB) was performed to examine myelination and neuronal degeneration within the central nervous system (CNS).

## Immunohistochemistry

Immunohistochemistry was performed on formalin-fixed, paraffin-embedded (FFPE) sections of cerebrum of affected animals (no. 1–12) and age-matched controls (no. 1009–1016) using monoclonal antibodies directed against 2′,3′-cyclic nucleotide-3′-phosphodiesterase (CNPase; MAB 326, dilution 1:100; Chemicon International), β-amyloid precursor protein (β-APP; MAB 348, dilution 1:800, Chemicon International), and polyclonal antibodies against myelin basic protein (MBP; AB 980, dilution 1:800, Chemicon International), neurite outgrowth inhibitor A (NogoA; AB5664, dilution 1:500, Merck Milipore), glial fibrillary acidic protein (GFAP; Z 0334, dilution 1:200, DAKO) and ionized calcium binding adaptor molecule 1 (Iba-1; PA5-27436, dilution 1:500, Thermo Fisher Scientific Inc.) to detect oligodendrocytes, axonal damage, astrocytes, and macrophages/microglia, respectively. Following a described standard protocol [38] endogenous peroxidase was blocked by a treatment with 0.5% $H_2O_2$ diluted in 80% ethanol. Subsequently, sections were heated in a sodium citrate buffer for antigen retrieval followed by blocking non-specific bindings with 20% goat serum. Thereafter sections were incubated with the primary antibodies for 90 minutes. Sections for negative control were incubated with non-immune serum. Biotinylated goat-anti-mouse IgG (BA-9200, dilution 1:200, Vector Laboratories) was used as secondary antibody. To visualize positive antigen-antibody-reaction, sections were incubated with avidin-biotin-peroxidase complex (ABC, PK-6100, Vector Laboratories) followed by 3,3′-diaminobenzidine tetrahydrochloride (DAB) with 0.03% $H_2O_2$, pH 7.2 for 5 minutes and slightly counterstained with Mayer's hemalaun. For polyclonal antibodies against Iba-1 the same protocol was used but the secondary antibody was replaced by biotinylated goat-anti-rabbit IgG (BA-1000, dilution 1:200, Vector Laboratories). For polyclonal antibodies against MBP and GFAP the same protocol was used without the heating step and using biotinylated goat-anti-rabbit IgG as secondary antibody.

For quantitative analysis of mature oligodendrocytes characterized by NogoA-immunohistochemistry, a morphometric grid (number of positive cell/0,0625mm$^2$) was used.

## Combination of immunohistochemistry and histochemistry

A combination of immunohistochemistry and histochemistry, using APP and LFB, was performed as described to point out the localization of damaged axons [38].

## Immunofluorescence

Immunofluorescence was performed to visualize the distribution and localization of TSEN54 in the cerebrum according to an established protocol [39]. For this purpose, a monoclonal mouse anti-human-TSEN54-antibody (anti-TSEN54, sc-398327, dilution 1:50, Santa Cruz Biotechnology, Inc) was used. Following the same procedure as described above, after dewaxing and antigen retrieval, non-specific binding was blocked with 20% goat serum diluted in diluted in PBS/0.1% Triton X/1% BSA. Thereafter the sections were incubated with the primary anti-TSEN54 antibody, diluted in PBS/0.1% Triton X/1% BSA. Sections for negative control were incubated with non-immune serum.

Goat-anti-mouse IgG CyTM3 (115-165-166, dilution 1:200, Jackson ImmunoResearch) was used as secondary antibody. Nuclear counter-staining was performed with 0.01%

bisbenzimide (Sigma-Aldrich Chemie GmbH) and sections were mounted with Dako fluorescence mounting medium (DakoCytomation GmbH).

## Statistical analysis

Statistical analysis of non-normal distributed data generated by NogoA-immunohistochemistry was performed by using IBM "Statistic Package for Social Sciences" SPSS program for Windows (version 24) and applying a Mann-Withney U-test for two independent samples. A *p*-value of less than or equal to 0.05 was considered statistically significant.

## Animal selection for genetic analysis

The genetic study included 381 Standard Schnauzers (S1 Table) from different European countries and the USA. Twelve closely related dogs presented clinically and/or histopathologically with leukodystrophy were designated as cases (no. 1–12). The remaining 369 Standard Schnauzers represented population controls, for which there were no records of neurological disease. A subset of 237 controls were collected in Finland. This subset was used to estimate the disease allele frequency in a representative population. Samples from 627 additional dogs of 72 genetically diverse breeds, which had been donated to the Vetsuisse Biobank, were also used as controls (S5 Table).

## DNA extraction

Genomic DNA was extracted from EDTA blood and hair samples according standard methods using the Maxwell RSC Whole Blood DNA and the Maxwell RSC Blood DNA Kits in combination with the Maxwell RSC machine (Promega). Genomic DNA from FFPE tissue samples was extracted using the Maxwell RSC DNA FFPE Kit according manufacturer's instructions. Furthermore, a semi-automated DNA extraction robot (PerkinElmer chemagen Technologie GmbH) was used to extract DNA from the Finnish Standard Schnauzer samples.

## Linkage analysis and homozygosity mapping

Linkage analysis and homozygosity mapping with two families were performed (Fig 8). Genotype data for 13 members of these two families (no. 1–4, 13–21) were obtained with Illumina CanineHD BeadChips by Geneseek/Neogen and used for a parametric linkage analysis (S1 File). For all dogs, the call rate was > 95%. Using PLINK v 1.07 [40], markers that were non-informative, located on the sex chromosomes, or missing in any of the 13 dogs, had Mendel errors, or a minor allele frequency < 0.05, were removed. The final pruned dataset contained 88,546 markers. To analyze the data for parametric linkage, an autosomal recessive inheritance model with full penetrance, a disease allele frequency of 0.5 and the Merlin software [41] were applied.

For homozygosity mapping, the genotype data for the four affected dogs were used. Markers that were missing in one of the four cases, markers on the sex chromosomes and markers with Mendel errors in the family were excluded. The --homozyg and --homozyg-group options in PLINK were used to search for extended regions of homozygosity > 1 Mb. The output intervals were matched against the intervals from linkage analysis in Excel spreadsheets to find overlapping regions.

## Whole genome resequencing and variant filtering

An Illumina TruSeq PCR-free library with an insert size of 400 bp was prepared from one affected Standard Schnauzer (SM010) and 193 million 2 x 150 bp paired-end reads were

obtained on an Illumina HiSeq 3000 instrument (21x coverage). Mapping and variant calling was done as described [42]. The sequence data were deposited under study accession PRJEB16012 and sample accession SAMEA104500412 at the European Nucleotide Archive. Functional effects and genomic context of the called variants were annotated using SnpEff software [43] together with the NCBI *Canis lupus familiaris* annotation release 104. For private variant filtering we used control genome sequences from 8 wolves and 213 dogs. These genomes were either publicly available [44] or produced during other previous projects (S3 Table).

### PCR and Sanger sequencing

We used a Sanger sequencing protocol for targeted genotyping of the TSEN54:c.371G>A variant. Specifically, a 589 bp PCR product was amplified from genomic DNA using the AmpliTaq-Gold360Mastermix (Life Technologies) or the Biotools DNA Polymerase (Biotools B&M Labs, S.A.) together with primers 5'-CGA AGA GGC CTT GTA TCT GC-3' (Primer F) and 5'-AAT TGC CAC CAC TAG GAT GC-3' (Primer R). For the FFPE samples, a 254 bp PCR product was amplified from genomic DNA using the AmpliTaqGold360Mastermix (Life Technologies) together with primers 5'-GCC-TGG-AAG-TTG-CTC-CTT-TA-3' (Primer FFPE F2) and 5'-CGG-CAC-CTC-ACT-GGT-ACT-C-3' (Primer FFPE R2). After treatment with exonuclease I and alkaline phosphatase, amplicons were sequenced on an ABI 3730 DNA Analyzer (Life Technologies). Sanger sequences were analyzed with the Sequencer 5.1 software (GeneCodes).

### Reference sequences

All analyses were performed using the CanFam 3.1 dog genome assembly as reference sequence. Numbering within the canine *TSEN54* gene refers to NCBI RefSeq accessions XM_540434.6 (mRNA) and XP_540434.3 (protein) unless explicitly stated otherwise. Numbering within the human *TSEN54* gene refers to NCBI RefSeq accessions NM_207346.2 (mRNA) and NP_997229.2 (protein).

## Supporting information

**S1 File. SNV microarray genotypes of 13 Standard Schnauzers (ped- and map-file).**
(ZIP)

**S1 Table. Overview of examined dogs and results.**
(XLSX)

**S2 Table. Linkage and homozygosity data.**
(XLSX)

**S3 Table. Public genome accessions of 214 dogs and 8 wolves.**
(XLSX)

**S4 Table. Private variants in the genome of the affected Standard Schnauzer no.2.**
(XLSX)

**S5 Table. *TSEN54*:c.371G>A genotypes of 627 dogs from 72 different dog breeds.**
(XLSX)

## Acknowledgments

We thank all the owners who donated samples and information on their dogs. Dr. Peter Dziallas, Dr. Beate Länger and Franziska Anders are acknowledged for excellent diagnostic imaging support. Additionally we thank Julia Baskas, Nathalie Besuchet Schmutz, Muriel Fragnière,

Petra Grünig, Reetta Hänninen, Claudia Hermann, Katharina Lange, Ileana Quintero, Sabrina Schenk and Caroline Schütz, for their excellent technical support. The Next Generation Sequencing Platform and the Interfaculty Bioinformatics Unit of the University of Bern performed the whole genome sequencing experiment, and provided high performance computing infrastructure. We thank the Dog Biomedical Variant Database Consortium (Gus Aguirre, Catherine André, Danika Bannasch, Doreen Becker, Brian Davis, Cord Drögemüller, Kari Ekenstedt, Kiterie Faller, Oliver Forman, Steve Friedenberg, Eva Furrow, Urs Giger, Christophe Hitte, Marjo Hytönen, Vidhya Jagannathan, Tosso Leeb, Hannes Lohi, Cathryn Mellersh, Jim Mickelson, Leonardo Murgiano, Anita Oberbauer, Sheila Schmutz, Jeffrey Schoenebeck, Kim Summers, Frank van Steenbeek, Claire Wade) for sharing whole genome sequencing data from control dogs. and wolves. We also acknowledge all canine researchers who deposited dog whole genome sequencing data into public databases.

## Author Contributions

**Conceptualization:** Wolfgang Baumgärtner, Andrea Tipold, Tosso Leeb.

**Data curation:** Vidhya Jagannathan.

**Funding acquisition:** Hannes Lohi, Wolfgang Baumgärtner, Andrea Tipold, Tosso Leeb.

**Investigation:** Theresa Störk, Jasmin Nessler, Linda Anderegg, Enrice Hünerfauth, Isabelle Schmutz, Vidhya Jagannathan, Kaisa Kyöstilä.

**Methodology:** Vidhya Jagannathan.

**Supervision:** Hannes Lohi, Wolfgang Baumgärtner, Andrea Tipold, Tosso Leeb.

**Visualization:** Theresa Störk, Jasmin Nessler, Tosso Leeb.

**Writing – original draft:** Theresa Störk, Jasmin Nessler, Linda Anderegg, Tosso Leeb.

**Writing – review & editing:** Theresa Störk, Jasmin Nessler, Linda Anderegg, Enrice Hünerfauth, Isabelle Schmutz, Vidhya Jagannathan, Kaisa Kyöstilä, Hannes Lohi, Wolfgang Baumgärtner, Andrea Tipold, Tosso Leeb.

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
