## [Decision Letter · Decision Letter 0]

7 Aug 2019

Dear Dr Leeb,

Thank you very much for submitting your Research Article entitled 'TSEN54 missense variant in Standard Schnauzers with leukodystrophy' to PLOS Genetics. Your manuscript was fully evaluated at the editorial level and by independent peer reviewers. The reviewers appreciated the attention to an important topic but identified some aspects of the manuscript that should be improved.

We therefore ask you to modify the manuscript according to the review recommendations before we can consider your manuscript for acceptance. Your revisions should address the specific points made by each reviewer.

[LINK]

Yours sincerely,

Gregory S. Barsh

Editor-in-Chief

PLOS Genetics

Gregory Copenhaver

Editor-in-Chief

PLOS Genetics

Reviewer's Responses to Questions

**Comments to the Authors:**

Reviewer #1: review is uploaded as an attachment.

Reviewer #2: The authors report a novel autosomal recessive trait in Standard Schnauzers characterized by myelin loss primarily in the centrum semiovale. Histologically, they demonstrate that the lesions are predominantly demyelination with gliosis, macrophage/microglia activation and some evidence of axonal damage, Using linkage and homozygousity mapping, they identified critical intervals for the causal allele. A whole genome sequence of an affected dog was filtered for private homozygous variants not found in control WGS predicted to alter protein function. A highly conserved variant in the TSEN54 gene was the only variant to fit those criteria. That variant was concordant with the disease phenotype in the families. It was not common in DNA from a wider sample of Standard Schnauzers and not found in a random sampling of DNA from other breeds. Immunohistochemistry did not demonstrate any difference in distribution or intensity of TSEN54 signal between affected and control dogs.

The data show a strong association between the variant and the disease phenotype. The authors, however, are unable to demonstrate a functional effect of the variant with the studies performed and the biological function of the protein does not provide a compelling explanation for the pathogenesis of the limited demyelination seen in the dogs. TSEN54 variants have been associated with a different neurodevelopmental phenotype in humans which could just represent species difference, but again does not provide compelling evidence for causality. The authors appropriately recognize that they cannot rule out the possibility that the variant is a tightly linked benign variant. None-the-less, this study should stimulate further study of the gene’s role in myelination and could lead to improved understanding of the maintenance of myelin during development. Even if it does prove to be an unrelated linked marker, a DNA test will allow breeders to use wise breeding strategies to reduce the risk of the disease in the breed.

**Have all data underlying the figures and results presented in the manuscript been provided?**

Reviewer #1: Yes

Reviewer #2: Yes

PLOS authors have the option to publish the peer review history of their article (what does this mean?). If published, this will include your full peer review and any attached files.

Reviewer #1: No

Reviewer #2: No

---

## [Editor Report · Decision Letter 1]

10 Sep 2019

Dear Dr Leeb,

We are pleased to inform you that your manuscript entitled "TSEN54 missense variant in Standard Schnauzers with leukodystrophy" has been editorially accepted for publication in PLOS Genetics. Congratulations!

Yours sincerely,

Gregory S. Barsh

Editor-in-Chief

PLOS Genetics

Gregory Copenhaver

Editor-in-Chief

PLOS Genetics

Comments from the reviewers (if applicable):

**Data Deposition**

http://datadryad.org/submit?journalID=pgenetics&manu=PGENETICS-D-19-00985R1

**Press Queries**

---

## [Editor Report · Acceptance letter]

27 Sep 2019

PGENETICS-D-19-00985R1 

TSEN54 missense variant in Standard Schnauzers with leukodystrophy 

Dear Dr Leeb, 

We are pleased to inform you that your manuscript entitled "TSEN54 missense variant in Standard Schnauzers with leukodystrophy" has been formally accepted for publication in PLOS Genetics! Your manuscript is now with our production department and you will be notified of the publication date in due course.

With kind regards,

Kaitlin Butler

PLOS Genetics

On behalf of:
